# Molecular Characterization and Comparative Genomics of IncQ-3 Plasmids Conferring Resistance to Various Antibiotics Isolated from a Wastewater Treatment Plant in Warsaw (Poland)

**DOI:** 10.3390/antibiotics9090613

**Published:** 2020-09-17

**Authors:** Marta Piotrowska, Lukasz Dziewit, Rafał Ostrowski, Cora Chmielowska, Magdalena Popowska

**Affiliations:** 1Department of Bacterial Physiology, Institute of Microbiology, Faculty of Biology, University of Warsaw, 02-096 Warsaw, Poland; m.piotrowska@biol.uw.edu.pl (M.P.); rafalostrowski@biol.uw.edu.pl (R.O.); 2Department of Environmental Microbiology and Biotechnology, Institute of Microbiology, Faculty of Biology, University of Warsaw, 02-096 Warsaw, Poland; ldziewit@biol.uw.edu.pl; 3Department of Bacterial Genetics, Institute of Microbiology, Faculty of Biology, University of Warsaw, 02-096 Warsaw, Poland; corachmiel@biol.uw.edu.pl

**Keywords:** plasmid, IncQ, antibiotic resistance, wastewater treatment plant

## Abstract

As small, mobilizable replicons with a broad host range, IncQ plasmids are widely distributed among clinical and environmental bacteria. They carry antibiotic resistance genes, and it has been shown that they confer resistance to β-lactams, fluoroquinolones, aminoglycosides, trimethoprim, sulphonamides, and tetracycline. The previously proposed classification system divides the plasmid group into four subgroups, i.e., IncQ-1, IncQ-2, IncQ-3, and IncQ-4. The last two subgroups have been poorly described so far. The aim of this study was to analyze five newly identified IncQ-3 plasmids isolated from a wastewater treatment plant in Poland and to compare them with all known plasmids belonging to the IncQ-3 subgroup whose sequences were retrieved from the NCBI database. The complete nucleotide sequences of the novel plasmids were annotated and bioinformatic analyses were performed, including identification of core genes and auxiliary genetic load. Furthermore, functional experiments testing plasmid mobility were carried out. Phylogenetic analysis based on three core genes (*repA*, *mobA*/*repB,* and *mobC*) revealed the presence of three main clusters of IncQ-3 replicons. Apart from having a highly conserved core, the analyzed IncQ-3 plasmids were vectors of antibiotic resistance genes, including (I) the *qnrS2* gene that encodes fluoroquinolone resistance and (II) β-lactam, trimethoprim, and aminoglycoside resistance genes within integron cassettes.

## 1. Introduction

In the natural environment, as a result of the presence of subminimal inhibitory concentrations of antibiotics, resistance genes encoded by mobile genetic elements, such as plasmids, transposons, and integrons, can be disseminated through horizontal gene transfer (HGT) [1,2,3,4,5,6]. Wastewater treatment plants (WWTPs) are the entry routes for antibiotic-resistant bacteria (ARBs), including pathogenic and multidrug-resistant (MDR) strains [7]. As sites where both antibiotics and favorable conditions for bacterial growth occur, WWTPs promote horizontal gene transfer (HGT) [8,9]. HGT events may lead to the rapid emergence and dissemination of antibiotic resistance among bacteria both in clinical and natural environments [10] and also to genetic exchange between clinical and WWTP strains [11,12]. WWTPs are a reservoir for bacteria harboring antibiotic resistance plasmids. Several studies showed the presence of ARGs in plasmid metagenomes of WWTPs and indicated that ARG profiles of activated sludge and final effluent are highly similar [13,14]. The occurrence of a similar gene pool at different stages of wastewater treatment is caused by the fact that antibiotics, especially fluoroquinolones, trimethoprim, or sulfonamides, are poorly removed during the wastewater treatment process. Generally, traditional WWTPs are not tailored for the removal of antibiotics or ARGs [15,16,17]. Multiple studies revealed the presence of ARGs and ARBs even in wastewater effluents [18,19,20,21], which poses a threat to their further dissemination to habitats downstream of the sewage plant [22]. The ARGs detected in WWTPs confer resistance to all major classes of antimicrobial drugs used both in medicine and veterinary medicine, including aminoglycosides, β-lactams, chloramphenicols, fluoroquinolones, macrolides, rifampicin, tetracyclines, trimethoprim, and sulphonamide [17]. Many of these resistance genes are located on resistance plasmids. Among these plasmids, there is a remarkable variety in size and load of genes either conferring resistance to antibiotics, detergents, heavy metals, or those encoding virulence factors [5].

Plasmids of the IncQ group are small-sized, mobilizable replicons with a broad host range (BHR). Their mechanism of replication relies on strand displacement [23]. This makes the plasmid DNA replication less dependent on the host cell apparatus, enabling their broad host range. An archetype of the IncQ group is the RSF1010 plasmid [24]. Since its identification, many other replicons of the Inc group have been discovered worldwide [25,26,27,28]. Previous studies indicated that an evolutionary success of IncQ plasmids in widespread dissemination is a result of their (a) small size, (b) high copy number, (c) ability to replicate in diverse bacterial hosts, and (d) ability to be transferred from one host to another. It was also shown that these plasmids carry various ARGs conferring resistance to aminoglycosides, β-lactams, carbapenem, chloramphenicol, florfenicol, lincosamides, sulphonamides, streptothricin, tetracyclines, and quinolones [29,30].

Loftie–Eaton and Rawlings [30] proposed to divide this group of plasmids into the following four subgroups: IncQ-1, IncQ-2, IncQ-3, and IncQ-4. Plasmids of the first two groups have been studied in detail compared to those of the IncQ-3 and IncQ-4 subgroups, to which little attention has been paid so far [29,31,32,33].

The main goal of this study was to perform a meta-analysis of five newly identified IncQ-3 plasmids in the light of general diversity of IncQ-3 replicons. The genomic structure, potential mobility, biogeography, and diversity of antibiotic resistance genes carried by these plasmids were investigated.

## 2. Results and Discussion

### 2.1. Structural and Functional Genomics of Five Novel IncQ Plasmids

In the course of our previous studies, a plethora of β-lactamase-resistant bacterial strains harboring various extrachromosomal replicons were identified [20,21]. Sequencing of their plasmidomes revealed a group of five small (7.6–9.6 kb) plasmids that were present in 16 clonally unrelated Gram-negative multireplicon strains. Based on an analysis of their replication systems, the plasmids were classified into the IncQ-3 subgroup. Interestingly, in all 16 strains various sets of plasmids accompanying IncQ-3 replicons were observed, which may exemplify HGT events among bacteria inhabiting the analyzed wastewater treatment plant, as observed previously for other environments [34,35,36]. Based on their genetic structure and gene load, the identified plasmids were divided into three groups: (i) p5.4_c4 and p115_p2, (ii) p458_p3 and p426_p3, (iii) p435_c4 (Figure 1, Table 1).

Plasmid p5.4_c4 was identified parallelly in two *Aeromonas* strains, i.e., 5.4 (GenBank: MF461069) and 6.45 (MF461079), which had been isolated from raw wastewater samples. Plasmid p115_p2 was identified in *Aeromonas* sp. 115, which had been found in raw wastewater and ten *Raoutella* sp. strains isolated from activated sludge samples, i.e., 210C (MF457856), 274B (MF457859), 213C (MF457857), 382A (MF457862), 293 (MF457865), 385A (MF457863), 299A (MF457866), 376 (MF457861), 286 (MF457864), and 328 (MF457860). Both replicons, i.e., p5.4_c4 and p115_p2, showed high similarity (sequence coverage of 84–100% and nucleotide sequence identity of 99–100%) to four other IncQ-3 plasmids, namely: pPCMI3 (MH569711) from clinical *Serratia marcescens* strain S89, pJF-789 (KX912254) from clinical *Klebsiella oxytoca* strain H140960789, pEMB2 (KJ631731) from an environmental, uncultured bacterium, and pQ7 (NC_014356) from clinical *Escherichia coli* strain 7 (Table 2).

The second group also contained two replicons, i.e., p458_p3 and p426_p3, which were identified within *Aeromonas* strains 458 (MF461156) and 426 (MF461154), respectively, both isolated from treated wastewater. For both replicons, 16 highly similar plasmid sequences were found in the NCBI database (coverage 70–100%, identity 97–100%), namely: pAHH04 (JN315883), pBRST7.6 (NC_011207), and pBF7.8 (KM245123), from three *Aeromonas hydrophila* strains isolated from infected fish, pAB5 (KU644674) from *Aeromonas caviae* isolated from a river, pFECR (MF554639) from an uncultured bacterium isolated from wastewater, pUR19829-KPC21 (MH133192) from clinical *E. coli* INSRA19829, pHP5 (KU644676) from an aquatic strain of *Aeromonas allosaccharophila*, pHP16 (KU644675) from an aquatic strain of *A. caviae*, pKPSH212 (KT896501), pKPSH70 (KT896500), pKPSH213.55 (KT896502), pKPSH231 (KT896503), and pKPSH169 (KT896499) from five wastewater strains of *K. pneumoniae*, pGNB2 (NC_013773) isolated from activated sludge, unnamed *Salmonella enterica* plasmid (MK191840), and pQnrS2_045523 (CP032896) from a clinical *Enterobacter kobei* strain WCHEK045523 (Table 2).

The last of the analyzed plasmids—p435_c4—was discovered in *Kluyvera* sp. 435 (MF457880) isolated from treated wastewater. This replicon was the most divergent from the other analyzed plasmids and showed the highest similarity to two plasmids, i.e., pJF-707 (KX946994) from clinical *K. oxytoca* H143640707 and pUL3AT (HE616889) isolated from *Enterobacter cloacae* LIM73 from hospital effluents (coverage 62%, identity 100%) (Table 2).

Comparative nucleotide analysis of the five newly identified plasmids indicated four core genes encoding: RepA (helicase), MobA-RepB (fusion gene of relaxase–primase), MobC (protein required for DNA cleavage), and a hypothetical protein. In all these plasmids, the *mobA* and *repB* genes were fused, which was previously observed as a characteristic feature of many plasmids within all IncQ groups. Previous studies did not clearly determine if the relaxase–primase gene fusion is expressed as two separate polypeptides or as a single peptide with two domains [43]. However, there seems to exist a strong selection for these fused transcripts among the IncQ plasmids, which suggests that both processes (i.e., replication and conjugation) may be coupled. Plasmids of the IncQ group possess replication systems operating via the strand-displacement mechanism. This system consists of three genes, *repBAC,* which form a cluster and an intergenic region of replication initiation (*oriV*) containing iterons, where the replication starts [44]. In plasmids analyzed in this study, three conserved direct repeats (DR)—putative iterons—were found within the *oriV* regions. The consensus iteron sequence is as follows: 5′-CCCCCACGGTAACTCNNCCC-3′. The NN represents an ambiguous dinucleotide: (i) CA in plasmids p435_c4, p458_p3, p426_p3, or (ii) TC in p5.4_c4 and p115_p2. Within each *oriV,* the following DNA regions were identified besides iterons: (a) G+C-rich region, (b) A+T-rich region, (c) 15-bp region, highly conserved in IncQ-like plasmids (5′-CTGCGCCTAGTGGAG-3′) and (d) two palindromic sequences (5′-CCGCGCCGAAGGGGCGCGG-3′ and 5′-ACCCCCGGAGGGGGT-3′). The *repB*, *repA*, and *repC* replication genes are highly similar in the identified plasmids (95–100% of nucleotide sequence identity). However, in plasmid p435_c4, no equivalent of the *repC* gene was identified, which suggests that the product of that gene may be delivered in trans or that the gene is unnecessary for plasmid replication. The latter hypothesis seems to be more probable, since the sequences of two other, highly similar IncQ-3 plasmids, also deprived of the *repC* gene, were found in the NCBI database, i.e., pJF-707 (KX946994) and pUL3AT (HE616889).

The genetic module responsible for the mobilization to conjugal transfer is also highly conserved within all identified plasmids. It consists of two genes—*mobA* and *mobC*. The *oriT* sites of the analyzed IncQ-3 plasmids were identified based on the homology to the previously identified *oriTs* [30] and all 29 plasmids contain the same, highly conserved, putative 12-bp region (5′-TTACACCTTGCT-3′) comprising an origin of transfer.

The analyzed plasmids were tested for their ability to be transferred into two hosts, *E. coli* DH5a (Rif^r^) and *Pseudmonas aeruginosa* PAO1161 (Rif^r^), via bi- and triparental conjugation (Appendix A). Interestingly, only two plasmids, p5.4_c4 and p115_p3, originally carried by *Aeromonas* spp. and *Raoultella* spp., were transferable, which is in agreement with previous findings concerning highly similar IncQ plasmids [28,29,38]. Transconjugants were ceftazidime-resistant, which also confirm that beta-lactamase genes carried by IncQ plasmids were truly expressed. In contrast, plasmids p426_p3, p458_p3, and p435_c4 carried by *Aeromonas* spp. and *Kluyvera* sp. were unable to be transferred via conjugation into the tested strains, although these plasmids were successfully introduced to both recipient strains by transformation (as shown in a control experiment). Therefore, we speculate that their mobilization systems may be inactive or that the plasmids need a mobilizing system different from the RK2 mating pair formation system (present in plasmid pRK2013) that was provided in the tri-parental mating.

Apart from the conserved backbone, the analyzed plasmids carry various sets of auxiliary genes, the majority of which are antibiotic resistance genes (Figure 1). The identified ARGs include β-lactam, quinolone, trimethoprim, and aminoglycoside resistance genes. The *bla*_FOX-15_ gene found in p435_c4 plasmid was previously identified as a new variant of *bla*_FOX_ gene encoding cephalosporin-hydrolyzing class C β-lactamase (MF795087) [21]. This gene was 96% identical at the level of nucleotide sequence to *bla*_FOX2_ from *E. coli* (NG_049102), *bla*_FOX3_ from *K. oxytoca* (NG_049103), and *bla*_FOX3_ from *Klebsiella pneumoniae* (LC072710). The sequence of the second β-lactamase gene *bla*_GES-7_ found in plasmids p5.4_c4 and p115_p2 was 100% identical to those of *bla*_GES-7_ from *S. marcescens* (AYD68573), *bla*_GES-7_ from *E. cloacae* subsp. *cloacae* (ANS91868), *bla*_GES-7_ from *Pseudomonas putida* (CZT31717), and *bla*_GES-7_ from *Aeromonas veronii* (ADU79013). Quinolone resistance (*qnrS2*) genes were found in p426_p3 and p458_p3. Identical gene sequences were found in numerous strains in the NCBI database, e.g., *E. coli* (MT219825), *A. caviae* (MN477222), or *Shewanella aestuarii* (CP050314). Trimethoprim resistance gene *dfrB3* found in p5.4_c4 and p115_p2 is another widespread ARG whose sequence shows 100% identity with that of *dfrB3* gene from *S. marcescens* (MH569711, CP020502), *Alcaligenes* sp. (KY047417), *Klebsiella aerogenes* (NG_047747), *E. coli* (KU997026), and other bacteria. The last ARG was aminoglycoside resistance gene *aac(6′)ib-cr* found in p115_p2. At the level of nucleotide sequence, the gene was identical with *aac(6′)ib-cr* from *K. pneumoniae* (NG_047288), *bla*_OXA_/*aac(6’)-Ib* from *E. coli* (FJ696404), and *bla*_OXA_/*aac(6’)-Ib* from *K. pneumoniae* (AY219651).

Other identified genetic modules of the analyzed plasmids include IS*4* family transposase *tnp* (plasmid p435_c4), ATPase subunits of an ABC transporter (p435_c4), type 3 integrases *intI3* (p5.4_c4 and p115_p2), and an aerotaxis sensor receptor protein (p458_p3) (Figure 1). A summary of the identified genes, including the size of the encoded proteins, their position, and predicted function is presented in Appendix A.

### 2.2. Comparative Genomics of Inc-Q3 Plasmids

As mentioned above, the clustering proposed by Loftie–Eaton and Rawlings (2012) divides IncQ plasmids into four subgroups: IncQ-1, IncQ-2, IncQ-3, and IncQ-4, based on (i) a high level of identity of their replication modules and (ii) significant diversity of their mobilization systems. In this study, we performed comparative analysis of all known IncQ-3 representatives. Based on our searches, as of 22 March 2020, the sequences of 29 putative IncQ-3 plasmids were present in the GenBank NCBI database, including five novel replicons, identified in the course of this study (Table 1). The pangenome of all investigated IncQ-3 plasmids is small and consists of 44 genes, 10 of which are ARGs. This fact was rather expected, taking into account small plasmid sizes (5.8–13.5 kbp). Despite their small size, most of the plasmids carried antibiotic resistance genes, which makes the IncQ-3 subgroup important vectors of antibiotic resistance spread. Within the pangenome, we identified 17 singletons, including *intI1* (integrase class 1, KX912254), *bla*_FOX-15_ (β-lactamase, MF795087), *bla*_KPC-21_ (carbapenemase, MH133192), *klcA*/*korC* anti-restriction system proteins, MH133192), 5 different transposase genes of the IS*4*, IS*30*, IS*5*, IS*1595*, and IS*66* families (WP_011191341, AXK00886, AUY12324, AUY12326, AUY12328, respectively), and 7 genes encoding hypothetical proteins (AIU93996, ARD69998, ARD69994, YP_002221300, AJS09376, AYL03488, CRY94842). The pangenome analysis revealed that the core genome of the analyzed IncQ-3 plasmids is composed of only three genes, i.e., *repA*, *mobA*/*repB,* and *mobC* (Figure 2b). The core genes form a conserved cluster that is present in all IncQ-3 replicons. These genes were used for the phylogenetic analysis (Figure 2a) that revealed the presence of three main clusters of the IncQ-3 plasmids (Figure 2).

The first cluster contains 16 plasmids, including p458_p3 and p426_p3. Apart from pAER-ba17, all plasmids in this cluster carry *qnrS2* gene, which encodes fluoroquinolone resistance. Plasmids in this cluster have diverse *oriV*s, differing in iteron number and sequence.

The second cluster contains three branches formed by nine plasmids, including p5.4_c4 and p115_p2 (Figure 2a). The first and second branch consist of three plasmids: pFECR, pAHH04, and pAB5, which carry *qnrS2* genes, similarly to those in cluster I. The third separate branch of cluster II consists of six replicons, which carry various antibiotic resistance genes (conferring resistances to β-lactams, trimethoprim, and aminoglycosides) that were located within class 1 or 3 integrons. Interestingly, each plasmid of the second branch of cluster II carries a variant of the *bla*_GES_ gene (β-lactamase).

The last cluster (III) consists of four plasmids divided into two branches. The former includes p435_c4, pJF-707, and pUL3AT and the latter contains one plasmid—pRGRH0378. Characteristic features of these plasmids are: (i) Differences in the structure of *oriV*s and (ii) the presence of variable set of ARGs, which makes them the most heterogeneous cluster. Furthermore, the lack of *repC* gene was observed within three plasmids in this cluster, i.e., p435_c4, pJF-707, and pUL3AT (Figure 2). Plasmid pRGRH0378 was obtained from a rat gut metagenome from hospital sewage in Denmark (LN853034.1). The plasmid possesses *repC* gene and all other core genes characteristic of the rest of IncQ-3a plasmids and only one additional gene encoding a hypothetical protein. That makes pRGRH0378 not only the smallest reported plasmid of the IncQ-3 group (5.8 kbp) but also the most conserved one.

### 2.3. IncQ-3 Plasmids as a Reservoir of Antibiotic Resistance Genes and Biogeography of IncQ-3 Plasmids

Comparative genomics of all 29 known IncQ-3 plasmids showed that 27 of them carry at least one antibiotic resistance gene that confers resistance to quinolone, β-lactams, or trimethoprim (Table 1 and Table 2). Two main groups of antibiotic resistance vectors were identified within investigated plasmids: (1) *qnrS2*-carrying plasmids that confer quinolone resistance, and (2) plasmids carrying β-lactam resistance genes—*bla*_GES_, *bla*_OXA_, and *bla*_FOX_—comprising integron gene cassettes (Figure 2c).

The *qnrS2*-carrying plasmid group consists of 18 replicons that have been found worldwide, mainly in wastewater [29,42] or other aqueous environments, including river water [28], hospital effluents [28], or infected fish [31,40]. These plasmids originated from diverse geographical regions, including Israel [42], Germany [29], China ([28], CP032896), Portugal (MH133192), USA (MK191840), Canada (MF554639), South Korea [40], and Poland (p458_p3 and p426_p3 found in this study) (Figure 3). The *qnrS2* genes belong to the most prevalent plasmid-mediated quinolone resistance (PMQR) gene group. Plasmid-mediated quinolone resistance was first identified in a *Klebsiella pneumoniae* clinical isolate from the United States in 1998 [45]. Since then, wide dissemination of *qnr* genes has been confirmed in both clinical and environmental strains. The dissemination is often associated with transposable elements (TEs) harbored by plasmids [46]. It was suggested that the *qnrS2* gene was potentially introduced into some of the IncQ-3 plasmids by transposition [28]. In seven of the analyzed plasmids, *qnrS2* is a part of a putative, nonautonomous, mobile insertion cassette, bordered by inverted repeats (IRs) and flanked by 5-bp long direct repeats (DRs). The insertion site in plasmids pAB5 [28] and pAHH04 [40] is different from that in the remaining counterparts, which suggests two independent transposition events.

One of the plasmids pUR19829-KPC21 (MH133192) additionally encodes a *bla*_KPC_ gene [41]. The carbapenemase gene is located on a putative complex mobile element, which is also present in pKP1194 plasmid identified in a clinical *K. pneumoniae* strain (KX756453) [41].

The plasmid-mediated *qnrS2* genes have been identified mainly in representatives of *Enterobacteriaceae*, *Shewanella* sp., *Vibrionaceae,* and *Aeromonas* spp. [47,48,49,50,51]. While the analyzed IncQ-3 plasmids were mostly isolated from *Aeromonas* spp. and *Klebsiella* spp., these replicons were also found in *E. coli*, *Salmonella enterica* sv. Typhimurium, and *E. kobei* (Table 2).

The second group consists of nine plasmids that carry β-lactamase genes from various groups: *bla*_GES_, *bla*_OXA_, and *bla*_FOX_. Simultaneously, most of these plasmids carry also other ARGs, i.e., *dfrB3* and *aac(6′)ib-cr,* encoding trimethoprim and aminoglycoside resistance, respectively. The analyzed plasmids were detected in clinical and wastewater samples in diverse geographical locations, i.e., Switzerland [32], Poland (MH569711.1, this study), South Korea [37], UK (KX946994), Luxemburg [39], and France [38]. Three of the plasmids in our study p5.4_c4, p115_p2, and p435_c4 carry *bla* genes also within an integron. Plasmids p5.4_c4 and p115_p2 possess *bla*_GES-7_ gene that encodes GES class A β-lactamase [52]. GES β-lactamases are known to be encoded by integron gene cassettes originating from clinical and environmental strains [53,54,55,56]. A majority of studies identified *bla*_GES_ genes within class 1 integrons. In contrast, within IncQ-3 plasmids, seven harbor class 3 integrons and only one contains a class 1 integron (Table 1 and Table 2). Integrons are frequently associated with mobile elements, such as transposons, which fosters their dissemination among bacteria. The class 3 integrons were possibly acquired by IncQ-3 plasmids through transposition, as previously suggested by Barraud et al. (2013) [38]. The putative mobile element is bordered by terminal IRs, which closely resemble IRs of the Tn*3* family, and is surrounded by 5-bp long DRs. Since no transposase gene was found, it is possible that the identified DNA region may constitute a nonautonomous transposable element, which may be mobilized by a functional transposase from another TE.

Horizontal transfer of integrons by nonautonomous TEs was previously proved experimentally in the case of integron mobilization units (IMUs), which formed a composite structure identified in the IncQ-1 family plasmid, pCHE-A [57]. The integron was surrounded by terminal IRs. Providing a Tn*3*-family transposase in trans resulted in transposition of the composite element into another plasmid (generating 5-bp long DRs). This indicates that nonautonomous TEs may play a role in spreading of integrons and—consequently—antibiotic resistance genes among the IncQ plasmids.

## 3. Materials and Methods

### 3.1. Isolation of IncQ-3 Plasmids

Five plasmids analyzed in this study were found in 16 strains originating from “Czajka” Wastewater Treatment Plant in Warsaw, Poland. The strains belong to *Aeromonas* spp. or *Enterobcteriaceae* (Appendix A). A detailed description of sample collection, bacterial isolation, sewage treatment processing, and the identified strains was provided in our previous papers [20,21]. Previously, we investigated 162 bacterial isolates exhibiting resistance to β-lactams. In 110 strains, at least one extrachromosomal replicon was found; however, a majority of strains shared common plasmid profiles, as judged from electrophoretic analysis. Thirty strains with unique plasmid profiles were selected for further analyses. Complete nucleotide sequences of their plasmids were determined from plasmids isolated in 2018. Among the sequenced plasmids, the most numerous ones were small (7.6–9.6 kb) IncQ-3 plasmids (Table 1). These plasmids have been thoroughly analyzed in this study.

### 3.2. DNA Sequencing and Assembly

The complete nucleotide sequences of the plasmids were determined using Illumina MiSeq (Illumina Inc., San Diego, CA, USA) and MinION (Oxford Nanopore, Oxford, UK) sequencers yielding around 100× sequence coverage for each plasmid. Total plasmid DNA of the strains was sequenced in the DNA Sequencing and Oligonucleotide Synthesis Laboratory (oligo.pl) at the Institute of Biochemistry and Biophysics, Polish Academy of Sciences (Warsaw, Poland). Raw reads obtained were assembled de novo into contigs and scaffolds using the Unicycler hybrid approach (Unicycler v. 0.4.6). Unicycler is open source (GPLv3) and available at github.com/rrwick/Unicycler. Final gap closure was performed by PCR amplification of DNA fragments, followed by Sanger sequencing with an ABI3730xl Genetic Analyzer (Applied Biosystems, Waltham, MA, USA).

### 3.3. Gene Annotation and Bioinformatics

The rapid annotation of the complete nucleotide sequences of plasmids was performed with on-line MAISEN tool [58]. Additional manual annotations and preparation of plasmid files for deposition were performed using Artemis software version 16.0.0 [59]. Comparative genomic analyses were carried out with BLASTn and BLASTx tools [60]. Comparative genomic data were visualized using EasyFig software version 2.2.4. [61]. Phylogenetic analysis was performed with MEGA7 [62]. Transposable elements were identified by comparative analyses (BLASTn, BLASTx) using ISfinder database [63].

### 3.4. Plasmid Mobility Testing

Mobility (via conjugation) of the identified IncQ-3 plasmids was determined in bi- and triparental mating assays. Before the experiment, minimal inhibitory concentrations (MICs) were determined for chosen antibiotics according to M07-A9 guidelines of the Clinical and Laboratory Standards Institute (Appendix A. Sixteen donor strains of the IncQ-3 plasmids and recipient strains were tested for plasmid-encoded resistance to the following drugs: Ceftazidime (CAZ; 128—0.5 μg/mL), trimethoprim (TRI; 512—1 μg/mL), ciprofloxacin (CIP; 1024—0.5 μg/mL), tetracycline (TET; 512—1 μg/mL), and rifampicin (RIF; 256—0.5 μg/mL). Rifampicin-resistant *E. coli* DH5αR (Rif^r^), tetracycline-resistant *E. coli* JCB816 (Tet^r^), and *Pseudomonas aeruginosa* PAO1161 (Rif^r^) were used as recipient strains. In triparental mating, *E. coli* strain DH5a with pRK2013 helper plasmid was additionally used [64]. Transconjugants were selected on Lysogeny Broth agar plates supplemented with appropriate antibiotics, as listed in Appendix A. Transconjugant identity was verified in a three-step assay: (1) Single colonies of potential transconjugants were plated on chromogenic coliform agar (Biomaxima, Poland) (*E. coli*—blue colonies, *Aeromonas* sp./*Raoultella* sp./*Kluyver*a sp.—pink colonies, *P. aeruginosa*—beige colonies); (2) PCR analysis with primers listed in Appendix A was performed using isolated plasmid DNA as a matrix; (3) plasmids from transconjugants were isolated following alkaline lysis procedure (Sambrook and Russell, 2001) and visualized by electrophoresis and DNA staining.

As a control of the conjugal transfer experiments, plasmid DNA was introduced into recipient strains via chemical transformation. Plasmid DNA was isolated from the donor strains using a classical alkaline lysis procedure [65] and introduced into *E. coli* DH5α and *P. aeruginosa* PAO1 by chemical transformation [66,67].

### 3.5. Plasmid Sequences Accession Numbers

The complete nucleotide sequences of plasmids were deposited in GenBank (NCBI) database under the accession numbers MT231818-MT231822 (Table 1). A complete annotation table is also available in Appendix A.

## 4. Conclusions

As the reservoirs of antibiotic-resistant bacteria, WWTPs are ideal places for horizontal gene transfer of various ARGs. IncQ plasmids belong to a group of widely distributed replicons, occurring mostly in aquatic environments, including wastewater treatment plants, hospital effluents, and clinical samples. Previously proposed classification system divides IncQ group of plasmids into four subgroups: IncQ-1, IncQ-2, IncQ-3, and IncQ-4. The aim of this study was to analyze IncQ-3 plasmids whose sequences had been deposited in the NCBI database, plus five newly identified replicons (in five *Aeromonas* sp., ten *Raoutella* sp., and one *Kluyvera* sp. strain) from the “Czajka” WWTP in Poland. Host strains for these novel plasmids were isolated from wastewater at various stages of its treatment: Raw wastewater, activated sludge, or treated wastewater.

Based on phylogenetic analysis of three plasmid core genes, the identified replicons were divided into three clusters. However, all these plasmids have highly conserved backbones. The majority of the studied plasmids (27 out of 29) were vectors of ARGs and collectively carried seven different genes that encode resistance to four groups of antibiotics (β-lactams, fluoroquinolones, trimethoprim, and aminoglycosides). IncQ-3 as vectors of antibiotic resistance genes were divided into two main groups: (I) Vectors of *qnrS2* gene that encodes fluoroquinolone resistance and (II) vectors of β-lactam, trimethoprim, and aminoglycoside resistance genes located within an integron.

## Figures and Tables

**Figure 1 antibiotics-09-00613-f001:**
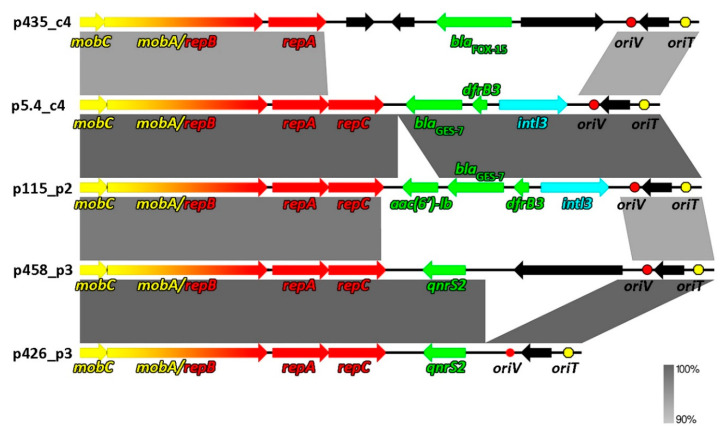
Linear maps showing the genetic structure of the circular IncQ-3 plasmids identified in the study. Genes assigned to specific modules are highlighted in different colors: Yellow—mobilization to plasmid transfer module, red—replication module, green—antibiotic resistance genes, blue—integrase. Red and yellow dots represent *oriV* and *oriT*, respectively. Comparative analyses were performed using 90% nucleotide sequence identity threshold (gray areas connecting DNA regions).

**Figure 2 antibiotics-09-00613-f002:**
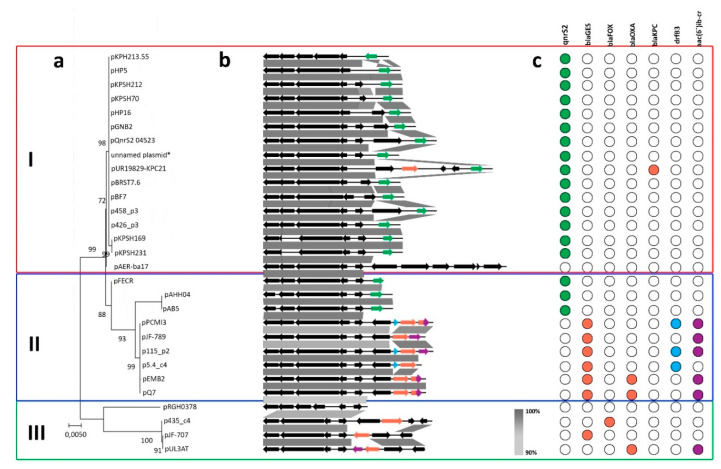
Comparative analyses of IncQ-3 plasmids: (**a**) Phylogenetic tree based on core genes of IncQ-3 replicons; (**b**) linear maps showing the schematic genetic structure of the circular IncQ-3 plasmids (retrieved from the NCBI database on 22 March 2020). The gray-shaded areas connect DNA regions of at least 90% sequence identity; ARGs were colored as on panel c. (**c**) ARGs carried by IncQ-3 plasmids conferring resistance to: Fluoroquinolones (green), β-lactams (orange), trimethoprim (blue), and aminoglycosides (violet).

**Figure 3 antibiotics-09-00613-f003:**
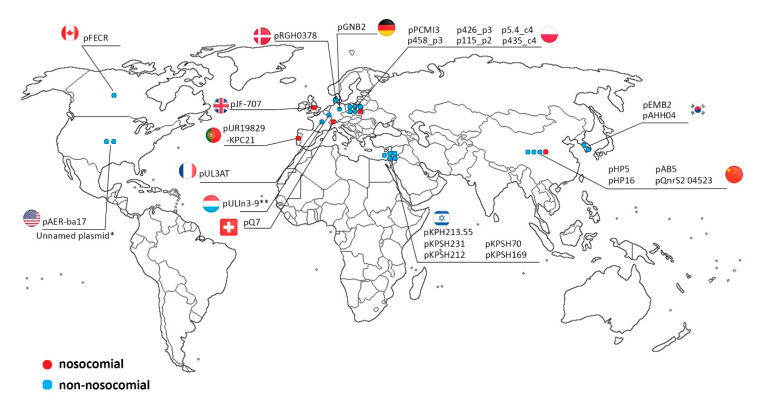
Biogeography of IncQ-3 plasmids: Blue dots—plasmids isolated from non-nosocomial environments; red dots—plasmids isolated from nosocomial environments.

**Table 1 antibiotics-09-00613-t001:** General information concerning the IncQ-3 plasmids identified in this study.

Plasmid Name	Plasmid Size (bp)	GC Content (%)	Host Strain (16S rDNA GenBank acc. no.)	Type of Wastewater	Antibiotic Resistance Genes	Integron Integrase Genes	GenBank Accession Number
p5.4_c4	8814	62	*Aeromonas* sp. (MF461069, MF461079)	RW	*bla*_GES-7_, *dfrB3*	*intI3*	MT231818
p115_p2	9448	61.5	*Aeromonas* sp. (MF461085); *Raoultella* sp. (MF457856- MF457866)	RW	*aac(6′)ib*, *bla*_GES-7_, *dfrB3*	*intI3*	MT231822
p458_p3	9639	57	*Aeromonas* sp. (MF461156)	TW	*qnrS2*	-	MT231819
p426_p3	7621	59.7	*Aeromonas* sp. (MF461154)	TW	*qnrS2*	-	MT231821
p435_c4	9407	61.2	*Kluyvera* sp. (MF457880)	AS	*bla* _FOX-15_	-	MT231820

Footnotes: RW—raw wastewater; AS—activated sludge; TW—treated wastewater.

**Table 2 antibiotics-09-00613-t002:** Summarized information concerning IncQ-3 plasmids retrieved from the GenBank (NCBI) database.

Plasmid	Size (bp)	Host Strain	Sample Origin	Antibiotic Resistance Genes	Integron Integrase Genes	GenBank Accession Number	Reference
pPCMI3	9448	*Serratia marcescens* subsp. *marcescens* S89	clinical sample, urine (Poland)	*blaOXA/aac(6’)-Ib-cr*, *bla*_GES-7_, *dfrB3*	*intI3*	MH569711	-
pJF-789	9016	*Klebsiella oxytoca* H140960789	clinical isolate	*bla*_GES-5_, *aac(6′)-lb*	*intI1*	KX912254	-
pEMB2	9042	uncultured bacterium	WWTP (South Korea)	*bla*_GES-1_, *bla*_OXA_/*aac(6′)-lb*	*intI3*	KJ631731	[37]
pQ7	9042	*Escherichia coli* 7	clinical sample (Switzerland)	*bla*_GES-1_, *bla*_OXA_/*aac(6′)-lb*	*intI3*	NC_014356	[32]
pJF-707	8300	*Klebsiella oxytoca* H143640707	clinical sample (UK)	*bla* _GES-5_	*intI3*	KX946994	-
pUL3AT	9005	*Enterobacter cloacae* LIM73	hospital effluent (France)	*bla*_OXA-10_, *aac(6’)-Ib*	*intI3*	HE616889	[38]
pULIn3-9		*Citrobacter freundii* LIM86	hospital effluent (Luxembourg)	*bla*_OXA-2_, *bla*_GES-1_	-	-	[39]
pAHH04	7191	*Aeromonas hydrophila* SNUFPC-A10	fish (South Korea)	*qnrS2*	-	JN315883	[40]
pAB5	7212	*Aeromonas caviae* AB5	river water (China)	*qnrS2*	-	KU644674	[28]
pFECR	6671	uncultured bacterium clone AA-102	WWTP (Canada)	*qnrS2*	-	MF554639	-
pUR19829-KPC21	12748	*Escherichia coli* INSRA19829	clinical sample (Portugal)	*qnrS2*	-	MH133192	[41]
pHP5	7637	*Aeromonas allosaccharophila* HP5	aquatic environment near hospital (China)	*qnrS2*	-	KU644676	[28]
pHP16	8213	*Aeromonas caviae* HP16	aquatic environment near hospital (China)	*qnrS2*	-	KU644675	[28]
pKPSH212	7742	*Klebsiella pneumoniae* I212	WWTP/ reclamation project site (Israel)	*qnrS2*	-	KT896501	[42]
pKPSH70	7748	*Klebsiella pneumoniae* I70	WWTP/ reclamation project site (Israel)	*qnrS2*	-	KT896500	[42]
pBRST7.6	7621	*Aeromonas hydrophila* AO1	infected fish	*qnrS2*	-	NC_011207	[31]
pGNB2	8469	-	activated sludge (Germany)	*qnrS2*	-	NC_013773	[29]
pKPSH213.55	6981	*Klebsiella pneumoniae* I213	WWTP/ reclamation project site (Israel)	*qnrS2*	-	KT896502	[42]
pKPSH231	7748	*Klebsiella pneumoniae* I231	WWTP/ reclamation project site (Israel)	*qnrS2*	-	KT896503	[42]
pKPSH169	7747	*Klebsiella pneumoniae* I169	WWTP/ reclamation project site (Israel)	*qnrS2*	-	KT896499	[42]
pBF7.8	7844	*Aeromonas hydrophila* IB101	infected fish	*qnrS2*	-	KM245123	-
unnamed	7555	*Salmonella enterica* subsp. *enterica* serovar Typhimurium var. 5- strain 63	swine (USA)	*qnrS2*	-	MK191840	-
pQnrS2_045523	9639	*Enterobacter kobei* WCHEK045523	clinical sample (China)	*qnrS2*	-	CP032896	-
pAER-ba17	13512	*Aeromonas* sp. ASNIH2	USA	-	-	CP026407	-
pRGRH0378	5790	uncultured prokaryote	from rat gut metagenome metamobilome, hospital sewer (Denmark)	-	-	LN853034	-

Footnotes: WWTP—wastewater treatment plant.

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
