# Peer review of "Molecular Characterization and Comparative Genomics of IncQ-3 Plasmids Conferring Resistance to Various Antibiotics Isolated from a Wastewater Treatment Plant in Warsaw (Poland)"

_antibiotics, 2020, doi:10.3390/antibiotics9090613_

Round 1

Reviewer 1 Report

The manuscripts herein submitted describe the charactrisation of environmental related plasmids carrying antibiotic resistance.They also analysed the horizontal gene transferring in E. coli and P. aeruginosa. 

By the way it could be interesting to also evaluate the activation and transcriptional evaluation of such gene class in different potential hosts.

Author Response

Manuscript ID: antibiotics-935281

Cover letter for reviewers

‘Molecular characterization and comparative genomics of IncQ-3 plasmids conferring resistance to various antibiotics isolated from a wastewater treatment plant in Warsaw (Poland)’

Marta Piotrowska, Lukasz Dziewit, Rafał Ostrowski, Cora Chmielowska, and Magdalena Popowska

Reviewer 1:

Comment: The manuscript herein submitted describe the characterisation of environmental related plasmids carrying antibiotic resistance. They also analysed the horizontal gene transferring in E. coli and P. aeruginosa. By the way it could be interesting to also evaluate the activation and transcriptional evaluation of such gene class in different potential hosts.

Answer: That is a very good point that transcriptional evaluation of transferred ARGs should be checked and it will add more information about functionality of these genes in different hosts. We decided to transfer our plasmids to E. coli and P. aeruginosa because these bacteria are frequently identified within WWTPs and other various aquatic environments. However, we think that transcriptional analysis within E. coli, P. aeruginosa and other bacterial hosts could be verified in further analysis. We decided to publish bioinformatics studies and basic functional analysis before this data will be out of date.

Reviewer 2 Report

This ms reports five IncQ-3 plasmids derived from a wastewater plant. Bioinformatic and experimental approaches were used. The findings provide new insights on IncQ-3 plasmids. Moreover, the study also presents data to further support the role of WWTPs as an important reservoir of resistance genes. The sequences of plasmids have been deposited into GenBank (Table 1; though not released yet). Comments are provided below:

The ms describes the comparison of the authors’ plasmids with those reported in literature. This comparison is welcome in the field.

Five IncQ plasmids were introduced to the tested recipients either by conjugation or transformation (L159-166). However, it is unclear what resistance phenotypes are conferred by these plasmids. Information in Table S4 is quite limited.  Host range of these plasmids is important, while resistance phenotypes are equally critical. Suggest authors to include relevant information in the main body of the ms to better understand the functionality of the plasmids. The latter may have been ignored by newer studies.

Tables 1 and 2. It will be very helpful to include the year of the plasmid isolation (some references may not have this type of information; if this is the case, the authors may even contact the authors of references for this information).

L18/30/53/etc. “beta”. Use Greek “β” as shown in L66 of the ms. Check this throughout the ms.

L35-37. The first para is one sentence only. It is better to merge with the second para.

L38/298. “antibiotic resistant bacteria” used twice in the ms with “ARBs” used once in L38. Suggest deleting “ARBs”.

L238. “5,8 kbp should be “5.8 kbp”. Please use “period” (English), not “comma” (French) as the “point”.

Author Response

Manuscript ID: antibiotics-935281

Cover letter for reviewers

‘Molecular characterization and comparative genomics of IncQ-3 plasmids conferring resistance to various antibiotics isolated from a wastewater treatment plant in Warsaw (Poland)’

Marta Piotrowska, Lukasz Dziewit, Rafał Ostrowski, Cora Chmielowska, and Magdalena Popowska

Reviewer 2:

  1. Comment: Five IncQ plasmids were introduced to the tested recipients either by conjugation or transformation (L159-166). However, it is unclear what resistance phenotypes are conferred by these plasmids. Information in Table S4 is quite limited.  Host range of these plasmids is important, while resistance phenotypes are equally critical. Suggest authors to include relevant information in the main body of the ms to better understand the functionality of the plasmids. The latter may have been ignored by newer studies.
  2. Answer: Please notice that all procedure how transconjugants were tested is described in Material and methods ‘Plasmid mobility testing’ section. It is said that ‘Transconjugants were selected on Lysogeny Broth agar plates supplemented with appropriate antibiotics, as listed in Table S1 (Supplementary material).’ In Table S1 there are listed antibiotics which were used to select transconjugants and next steps of verifying. However, we add a clarification of this fact to L162-164.

    2. Comment: Tables 1 and 2. It will be very helpful to include the year of the plasmid isolation (some references may not have this type of information; if this is the case, the authors may even contact the authors of references for this information).
  3. Answer: We added year of isolation of our plasmids into the ‘Material and methods’ section at L320. However, in case of plasmids from Table 2 this information is hard to obtain within this short amount of time or even impossible to gather. We can add section with ‘Year of plasmid deposition’ which is easy to obtain from GenBank, however we are not sure is it is a crucial information for this manuscript.

  4. Comment L18/30/53/etc. “beta”. Use Greek “β” as shown in L66 of the ms. Check this throughout the ms.
  5. Answer: ‘beta’ has been corrected to ‘β’ throughout the all manuscript.

    4. Comment L35-37. The first para is one sentence only. It is better to merge with the second para.
  6. Answer: Both para have been merged.

    5. Comment L38/298. “antibiotic resistant bacteria” used twice in the ms with “ARBs” used once in L38. Suggest deleting “ARBs”.
  7. Answer: At line L298 ‘ARGs’ abbreviation for ‘antibiotic resistance genes’ has been used, so it was not deleted since it is a different word. However, ‘ARBs’ abbreviation has been checked and corrected at L49 (‘s’ added).

    6. Comment L238. “5,8 kbp should be “5.8 kbp”. Please use “period” (English), not “comma” (French) as the “point”.
  8. Answer: Corrected to English ‘period’.

Reviewer 3 Report

The manuscript on molecular characterization of IncQ-3 plasmids from WWTP is well written and shares interesting facts. Plasmids along with other mobile elements works as a vector in the rise of ARBs. The IncQ plasmids have a wide host range and it is always helpful to have a detailed characterization of them . The authors have characterized and compared their  plasmids from WWTP to others of IncQ3 subgroup reported earlier with the aim to add additional insight to the subgroup.

Minor comments:

Line 174: check for misplaced punctuation .

Line 292: Reference citation requires editing.

Author Response

Manuscript ID: antibiotics-935281

Cover letter for reviewers

‘Molecular characterization and comparative genomics of IncQ-3 plasmids conferring resistance to various antibiotics isolated from a wastewater treatment plant in Warsaw (Poland)’

Marta Piotrowska, Lukasz Dziewit, Rafał Ostrowski, Cora Chmielowska, and Magdalena Popowska

Reviewer 3:

Minor comments:

1. Comment: Line 174: check for misplaced punctuation.

  1. Answer: Question mark has been deleted.

    2. Comment: Line 292: Reference citation requires editing.
  2. Answer: Reference citation has been updated.